# Causal Structural Hypothesis Testing and Data Generation Models

**Jeffrey Jiang**[*]
jimmery@ucla.edu

**Omead Pooladzandi**[*]
opooladz@ucla.edu

**Sunay Bhat**[*]
sunaybhat1@ucla.edu

**Gregory Pottie**[*]
pottie@ee.ucla.edu

## Abstract

A vast amount of expert and domain knowledge is captured by causal structural priors, yet there has been little research on testing such priors for generalization and data synthesis purposes. We propose a novel model architecture, Causal Structural Hypothesis Testing, that can use nonparametric, structural causal knowledge and approximate a causal model's functional relationships using deep neural networks. We use these architectures for comparing structural priors, akin to hypothesis testing, using a deliberate (non-random) split of training and testing data. Extensive simulations demonstrate the effectiveness of out-of-distribution generalization error as a proxy for causal structural prior hypothesis testing and offers a statistical baseline for interpreting results. We show that the variational version of the architecture, Causal Structural Variational Hypothesis Testing can improve performance in low SNR regimes. Due to the simplicity and low parameter count of the models, practitioners can test and compare structural prior hypotheses on small dataset and use the priors with the best generalization capacity to synthesize much larger, causally-informed datasets. Finally, we validate our methods on a synthetic pendulum dataset, and show a use-case on a real-world trauma surgery ground-level falls dataset. Our code is available on GitHub.[2]

## 1   Introduction

In most scientific fields, causal information is considered an invaluable prior with strong generalization properties and is the product of experimental intervention or domain expertise. These priors can be in a structural causal model (SCM) form that instantiates unidirectional relationships between variables using a Directed Acyclic Graph (DAG) [1]. The confidence in causal models needs to be higher than in a statistical model, as its relationships are invariant and preserved outside the data domain. In fields such as medicine or economics, where ground truth is often unavailable, domain experts are relied on to hypothesize and test causal models using experiments or observational data.

Generative models have been crucial to solving many problems in modern machine learning [2] and generating useful synthetic datasets. Causal generative models learn or use causal information for generating data, producing more interpretable results, and tackling biased datasets [3–5]. Recently, [6] introduces a *Causal Layer*, which allows for direct interventions to generate images outside the distribution of the training dataset in its CausalVAE framework. Another method, Causal Counterfactual Generative Modeling (CCGM), in which exogeneity priors are included, extends the counterfactual modeling capabilities to test alternative structures and "de-bias" datasets [7].

---

[*]Equal Contribution, Department of Electrical and Computer Engineering, University of California, Los Angeles
[2]https://github.com/SunayBhat1/Causal-Structural-Hypothesis-Testing

NeurIPS 2022 Workshop on Synthetic Data for Empowering ML Research.

CausalVAE and CCGM focus on causal discovery concurrently with simulation (i.e. reconstruction error-based training). But in many real-world applications, a causal model is available or readily hypothesized. It is often of interest to test various causal model hypotheses not only for in-distribution (ID) test data performance, but for generalization to out-of-distribution (OOD) test data. Thus we propose CSHTEST and CSVHTEST, which are causally constrained architectures that forgo structural causal discovery (but not the functional approximation) for causal hypothesis testing. Combined with comprehensive non-random dataset splits to test generalization to non-overlapping distributions, we allow for a systematic way to test structural causal hypotheses and use those models to generate synthetic data outside training distributions.

## 2 Background

### 2.1 Causality and Model Hypothesis Testing

Causality literature has detailed the benefits of interventions and counterfactual modeling once a causal model is known. Given a structural prior, a causal model can tell us what parameters are identifiable from observational data alone, subject to a no-confounders and conditioning criterion determined by d-separation rules [1]. Because the structural priors are not known to be ground truth, we assume a more deterministic functional form and we can make no assumptions about identifiability [8]. Instead, we rely on deep neural networks to approximate the functional relationships and use empirical results to demonstrate the reliability of this method to compare structural hypotheses in low-data environments.

Structural causal priors are primarily about the ordering and absence of connections between variables. It is the absence of a certain edge that prevents information flow, reducing the likelihood that spurious connections are learned within the training dataset distribution. Thus, when comparing our architecture to traditional deep learning prediction and generative models, we show how hypothesized causal models might perform worse when testing within the same distribution as the training data, but drastically improve generalization performance when splitting the test and train distributions to have less overlap. This effect is seen the most in small datasets where traditional deep learning methods, absent causal priors, can "memorize" spurious patterns in the data and vastly overfit the training distribution [9].

Our architectures explore the use of the causal layer, provided with priors, as a hypothesis-testing space. Both CSHTEST and CSVHTEST accept non-parametric (structural only, no functional-form or parameters) causal priors as a binary Structural Causal Model (SCM) and use deep learning to approximate the functional relationships that minimize a means-squared reconstruction error (MSE). Our empirical results show the benefits of testing structural priors using these architectures to establish a baseline for comparison where stronger causal assumptions cannot be satisfied.

## 3 Causal Hypothesis Gen and Variational Model

### 3.1 Causal Hypothesis Testing with CSHTEST

Our model CSHTEST, uses a similar causal layer as in both CCGM and CausalVAE [6, 7]. The causal layer consists of a structural prior matrix $\mathbf{S}$ followed by non-linear functions defined by MLPs. We define the structural prior $\mathbf{S} \in \{0,1\}^{d \times d}$ so that $\mathbf{S}$ is the sum of a DAG term and a diagonal term:

$$\mathbf{S} = \underbrace{\mathbf{A}}_{\text{DAG}} + \underbrace{\mathbf{D}}_{\text{diag.}} \tag{1}$$

$\mathbf{A}$ represents a DAG adjacency matrix, usually referred to as the causal structural model in literature, and $\mathbf{D}$ has 1 on the diagonal for exogenous variables and 0 if endogenous. Then, given tabular inputs $\boldsymbol{x} \in \mathbb{R}^d$, $\mathbf{S}_{ij}$ is an indicator determining whether variable $i$ is a parent of variable $j$.

From the structural prior $\mathbf{S}$, each of the input variables is "selected" to be parents of output variables through a Hadamard product with the features $\boldsymbol{x}$. For each output variable, its parents are passed through a non-linear $\eta$ fully connected neural-network. The $\eta$ networks are trained as general function approximators, learning to approximate the relationships between parent & child nodes:

$$\hat{\boldsymbol{x}}_i = \eta_i(\mathbf{S}_i \circ \boldsymbol{x}) \tag{2}$$

where $\mathbf{S}_i$ represents the $i$-th column vector of $\mathbf{A}$, and $\hat{\boldsymbol{x}}_i$ is the $i$-th reconstructed output [10]. In the case of exogenous variable $\boldsymbol{x}_i$, a corresponding 1 at $\mathbf{D}_{ii}$, 'leaks' the variable through, encouraging $\eta$ to learn the identity function while a 0 value forces the network to learn some functional relationship of its parents. The end-to-end structure, as seen in Figure 1, is trained on a reconstruction loss, defined by $\ell(\boldsymbol{x}, \hat{\boldsymbol{x}})$. We use the L2 loss (Mean Squared Error):

$$\ell_{\text{CSHTEST}} = ||\boldsymbol{x} - \eta_i(\mathbf{S}_i \circ \boldsymbol{x})||_2^2 \tag{3}$$

CSHTEST can be used, then, to operate as a structural hypothesis test mechanism for two structural causal models $\mathbf{S}$ and $\mathbf{T}$. The basic idea is that if $\ell_{\mathbf{S}} < \ell_{\mathbf{T}}$, across the majority of non-random OOD dataset splits for training and testing, then $\mathbf{S}$ is a more suitable hypothesis for the true causal structure of the data than $\mathbf{T}$. In section 4.3 we demonstrate the ID, OOD train/test splits to test this generalization capacity, and our experimental results provide baselines for this approach.

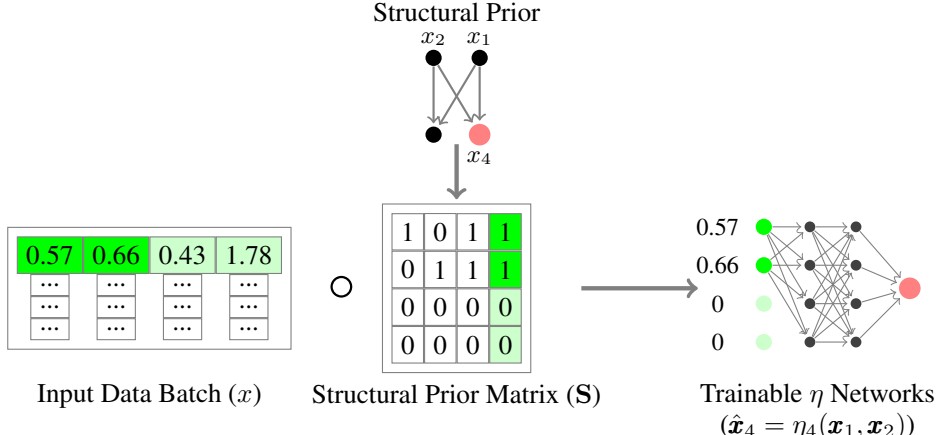

Figure 1: Causal Hypothesis Generative Architecture (CSHTEST) with an example of how the Structural Prior Matrix selects for the parents of each variable or identity if it is exogenous. The $\eta$ networks approximate the functional relationships in training.

### 3.2 Causal Variational Hypothesis Testing with CSVHTEST

We extend CSHTEST to a variational model CSVHTEST, that includes sampling functionality like a VAE [2]. We do this primarily for a more robust model in low Signal-to-Noise (SNR) regimes and to generate new data points that are not deterministic on the inputs, allowing for more dynamic synthetic data generation. CSVHTEST consists of an encoder, a CSHTEST causal layer and a decoder. Further details are provided in the appendix A.3.1.

## 4 Problem Setting

### 4.1 Structural Hamming Distance

In causal and graph discovery literature, the Structural Hamming Distance is a common metric to differentiate causal models by the number of edge modifications (flips in a binary matrix) to transform one graph to another [11, 12], often described as the norm of the difference between adjacency matrices:

$$\mathbf{H} = |\mathbf{A}^i - \mathbf{A}^j|_1 \tag{4}$$

However, Structural Hamming Distance does not account for the "causal asymmetry." The absence of edges is a more profound statement than inclusion, as any edge could have a weight of zero. Hence we define two types of hypotheses that are incorrect relative to ground truth, which could have the same Structural Hamming Distances:

- *Leaky* hypotheses are causal hypotheses with extra links. In general, having a leaky hypothesis will produce models that are more prone to overfitting, but with proper weighting, the solution space of a leaky causal hypothesis includes the ground truth causal structure.

- *Lossy* hypotheses are causal hypotheses where we are missing at least one link. Lossy hypotheses are much easier to detect because a lossy hypothesis results in lost information. As such, a lossy hypothesis should never do better than the true hypothesis, within finite sampling and noise errors.

From these definitions, we define the *Positive Structural Hamming Distance* and the *Negative Structural Hamming Distance*. We define these as, for null hypothesis $\mathbf{A}^0$ and alternative $\mathbf{A}^1$,

$$\mathbf{H}^+(\mathbf{A}^1, \mathbf{A}^0) = |\mathbf{A}^1 > \mathbf{A}^0|_1 \qquad \mathbf{H}^-(\mathbf{A}^1, \mathbf{A}^0) = |\mathbf{A}^1 < \mathbf{A}^0|_1 \tag{5}$$

where $\mathbf{H}^+$ counts how *leaky* the alternative hypothesis is and $\mathbf{H}^-$ counts how *lossy* it is. One remark is that $\mathbf{H} = \mathbf{H}^+ + \mathbf{H}^-$, but the "net" Hamming Distance $\Delta\mathbf{H} = \mathbf{H}^+ - \mathbf{H}^-$ can also be a naïve indicator of how much information is passed through the causal layer.

## 4.2 Baseline Models

### 4.2.1 Simulated DAG Baselines

We empirically test our theory that an incorrect hypothesis will result in worse OOD test error using extensive simulations. We use the same methodology as [13], simulating across multiple DAG nodes sizes, edge counts, OOD variable splits (described further in 4.3), and Structural Hamming Distance with iterations at the ground truth and modified DAG levels for robustness. In our experimental results, we calculate the probability a $\mathbf{H}$ of 1 closer to ground truth would have a lower OOD test error as the ratio across our simulations:

$$\Pr(\ell_{\text{CSHTEST}}(S_j) < \ell_{\text{CSHTEST}}(S_i)) \,\Big|\, 1 = |\mathbf{A}^i - \mathbf{A}_{GT}|_1 - |\mathbf{A}^j - \mathbf{A}_{GT}|_1 \tag{6}$$

where *GT* is ground truth, and so on for differences 2 and 3. In practice, we actually consider the probability conditional on a tuple of the positive and negative Hamming distances $(\mathbf{H}^+, \mathbf{H}^-)$ thus allowing us to distinguish hypotheses that are *leakier*, *lossier*, or the specific mix of the two. Doing so allows us to better consider the fundamental asymmetry in causality. Full hyperparameters and test cases can be found in Appendix A.5.

### 4.2.2 Sun Pendulum Image Dataset

A synthetic pendulum image dataset is introduced in [6] and we use it here to produce a physics-based tabular dataset where we know the ground truth DAG and can test the abilities of CSHTEST and CSVHTEST. More about the dataset is described in Appendix A.2.1.

### 4.2.3 Medical Trauma Dataset

We also analyze our model on a real-world dataset of brain-trauma ground-level fall patients that includes multiple health factors, with a focus on predicting a decision to proceed with surgery or not. We used an initial SHAP analysis to select three variables of high prediction impact: Glasgow Coma Scale/Score for head trauma severity (GCS), Diastolic Blood Pressure (DBP), the presence of any Co-Morbidities (Co-Morb), one demographic variable Age, along with the Surgery outcome of interest. Without the ground truth, we test two structural models shown in 2 based on knowledge of the selected variables and how they may interact to inform the surgery decision.

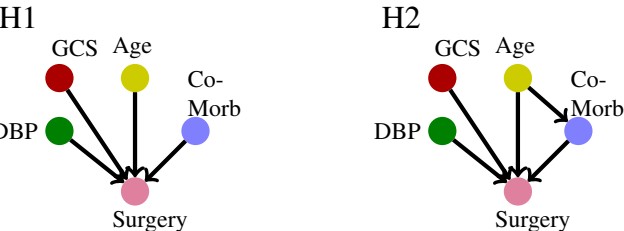

Figure 2: Two hypothesized structural causal priors for a medical dataset on trauma patients and the decision to perform surgery, H1 and H2.

### 4.3 Train/Test Data Splits

In order to test generalization error, we use a deliberate non-random split of our datasets (as well as a baseline random split). This is done on a single feature column of the tabular data at a time, splitting the data on that column at either the 25% or 75% quantile, with the larger side (either the upper or lower 75%) becoming the training data. An example of this train test split is visualized for both datasets in 3. We recommend viewing OOD test error across as many dimensions and split quantiles as possible given the size of the dataset and the available compute.

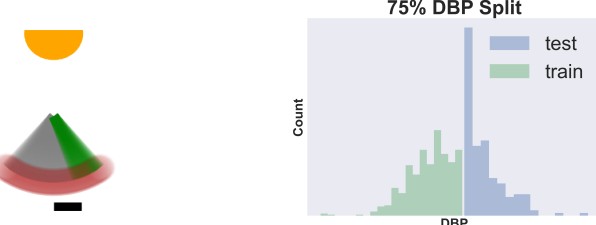

Figure 3: a) A 75% data-split on the pendulum angle feature (grey is training angle, green is testing angles b) A 75% data-split on the Diastolic Blood Pressure data.

## 5 Experiments

We test CSHTEST and CSVHTEST in multiple settings. First, we justify their usage by comparing their performance on both ID and OOD validation to their non-causal counterparts, showing that they operate as normal when trained in ID but perform much better when trained in OOD. We provide a table of relative loss probabilities to help interpret results using extensive simulations. Next, we observe the benefits and limitations of the CSHTEST method when we hypothesize several possible causal structures on the pendulum problem. Finally, we hypothesize and compare to structural priors on the medical dataset, and simulate new data.

### 5.1 Generalization Ability of CSHTEST

| | Pendulum Comparison | | | |
| | Random | | Split | |
| Method | Train | Test | Train | Test |
|---|---|---|---|---|
| NN | 0.02 | 0.02 | 0.04 | 10.27 |
| VAE | 0.11 | 0.06 | 16.97 | 89.4 |
| **CSHTEST** | 0.03 | 0.03 | 0.02 | 0.26 |
| **CSVHTEST** | 0.064 | 0.51 | 19.81 | 38.62 |

Table 1: Comparison of Traditional Deep Learning Techniques on a random and deliberate dataset split with CSHTEST and CSVHTEST when the ground truth causal structural information is known.

We compare the CSHTEST with a similarly sized fully-connected NN and CSVHTEST with a similarly sized VAE. The CSVHTEST also has the same causal layers as the CSHTEST so the variational models are larger overall than the CSHTEST and NN. Results of the pendulum are shown in Table 1. Against ID (random) data, the CSHTEST and CSVHTEST effectively perform the same, suggesting that there is no loss in representation by including the Causal Layer.

However, the CSHTEST and CSVHTEST models generalize much better to OOD data validation than their respective non-causal comparisons. This demonstrates the use of the CSHTEST and CSVHTEST as causal replacements to the NN and VAE. Conversely, as the out-of-distribution error rate can determine the "close-ness" of our model to the true causal model, this would enable the use of the OOD loss as a proxy for causal hypothesis testing.

### 5.2 Simulated DAG Hypothesis Testing

Figure 4 shows the type of empirical probability tables we can construct by simulating DAGs of various sizes under numerous conditions detailed in the appendix A.5. We note how by compar-

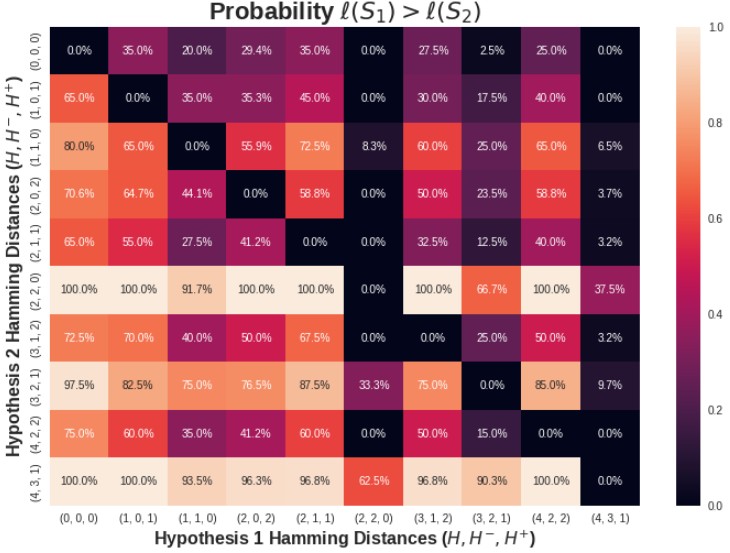

Figure 4: Probability table for a 4 node 4 edge DAG size with a linear SEM ground truth model for DAG simulations comparing hypothesis with various Hamming Distance Tuples.

ing the Hamming Distance tuples, we do not see a smooth gradient, but jumps as the *leaky/lossy* asymmetry is realized. Instead, by also incorporating the net Hamming distance to account for the causal asymmetry, we can explain the jumps. For instance, (2,2,0) shows a marked drop off compared to any lower Hamming Distance model — like ground truth (0,0,0) — because it has two edge losses ($\Delta \mathbf{H} = -2$) which vastly decreases needed information flow. In general, the upper triangle of this matrix should be below 50% and decrease as the Hamming Distances grow and get *lossier*. Within each Hamming distance, the values typically increase as $\Delta \mathbf{H}$ increases. Extensive simulations like this, done with similar assumptions to a comparable real-world problem, can provide baseline probabilities even if ground truth is not known, based on the relative Hamming Distances of the hypotheses. Further results DAG size 5x5 can be found in the appendix A.1.

### 5.3 Pendulum Hypothesis Testing

We consider 6 different hypotheses, shown in Figure 9, detailed in Appendix A.2.2. We arbitrarily do a 75% OOD split of the pendulum dataset on the sun position (as an exogenous example) and the shadow position (as an endogenous example) to test causal hypotheses.

The pendulum results are shown in Figure 5. We can clearly distinguish two tiers of results. One tier contains *GT*, *leaky*, and *2leaky*. This tier has a common $\mathbf{H}^- = 0$. All other DAGs, having $\mathbf{H}^- > 0$ show a very clear drop in OOD test performance. Thus, for hypothesis testing, we are able to distinguish causal hypotheses that are missing paths from ground truth. We leave it to further research to explore how to compare many hypotheses that achieve a similar loss, such as a criterion that favors the minimal hypothesized DAG.

Of interest is the *leak-loss* model, which has $\Delta \mathbf{H} = 0$. Its loss is generally lower than the purely lossy hypotheses, but still achieves a higher loss than the ground truth, despite graphically being the same level of connectedness. This result has the interesting consequence of CSHTEST being able to reject causal hypotheses with zero net Hamming distance.

### 5.4 Medical Data Hypothesis Testing

In the medical dataset, the second hypothesis from 2, which includes a path from Age to No-Comorbidities generalizes better than without the path, suggesting it is a better causal model. We use both trained architectures to simulate OOD data, with the causal models producing higher fidelity results to what we expect ground truth to be based on a holdout testset over the same distribution 6.

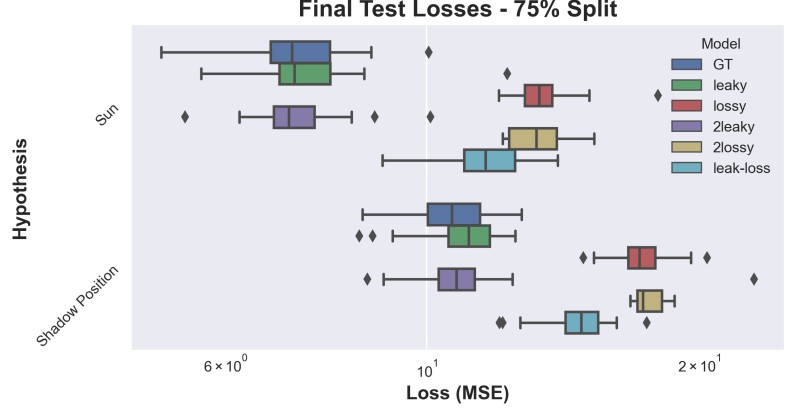

Figure 5: Final OOD Test Error Rates of Each Hypothesized DAG structure in the Pendulum Problem over Two Splits. See Appendix A.2.4 for numerical values and training trajectories.

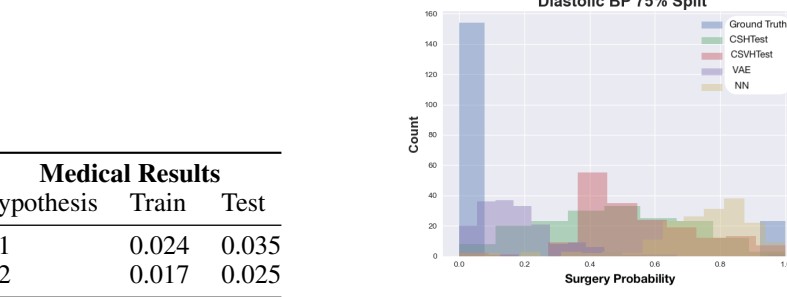

| Medical Results | | |
|---|---|---|
| Hypothesis | Train | Test |
| H1 | 0.024 | 0.035 |
| H2 | 0.017 | 0.025 |

(a) Medical results of CSHTEST for each of the hypothesized causal models.

(b) The test dataset reconstruction distributions for all models using medical H1 on the Diastolic BP 75% data

Figure 6: Medical Dataset Results

## 6  Conclusion

In this paper, we demonstrate the value of CSHTEST and CSVHTEST as causal model hypothesis testing spaces and the implications as generative models. We verify the effectiveness of our methodology on extensive simulated DAGs where ground truth is known, and we further show performance with ground truth and incorrect causal priors on a physics-inspired example. We show how CSHT-EST can be used to test causal hypotheses using a real-world medical dataset with ground level fall, trauma surgery decisions. CSHTEST offers a novel architecture, along with a deliberate data split methodology that can empower practitioners and domain experts to improve causally informed modeling and deep learning. There is extensive further research needed to fully realize the utility of structural causal hypothesis testing in conjugation with deep learning function approximation. We hope to better differentiate leaky causal models, without constraints on losses, using minimum entropy properties such as in [14] . We also hope to extend both CSHTEST and CSVHTEST to more flexible architecture which can combine recent progress with differential causal inference and binary sampling to better automate full or partial causal discovery. The results are a promising start to much further research integrating deep learning causal models with real-world priors and domain knowledge.

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

# A    Appendix

## A.1    DAG Simulation Results

### A.1.1    DAG Size 5x5: Linear

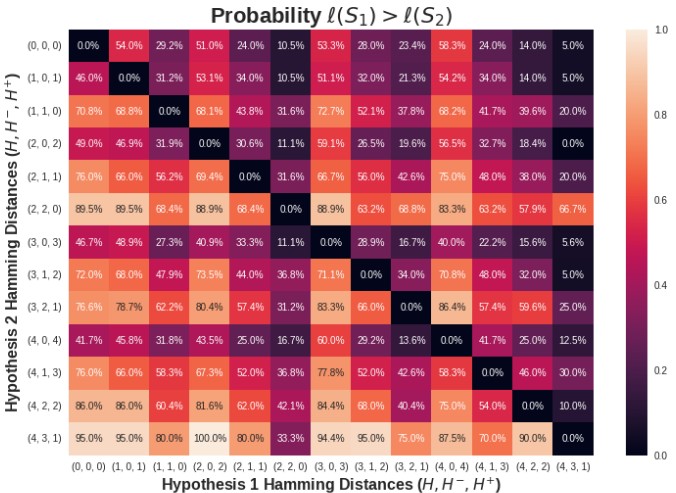

Figure 7: Probability table for a 5 node 5 edge DAG size with a linear SEM ground truth model for DAG simulations comparing hypothesis with various Hamming Distance Tuples.

## A.2    More about the Pendulum

### A.2.1    Pendulum Dataset

This dataset is generated by sweeping sun positions ($x_{sun}$) and pendulum angles ($\theta$) to produce realistic shadow width ($w_{shadow}$) and shadow locations ($x_{shadow}$) from deterministic non-linear functions. Figure 8 shows the true DAG and an example generated image. Here, the sun and pendulum variables are exogenous, and the shadow variables are endogenous. We take these values $\mathbf{u} = [\theta, x_{sun}, w_{shadow}, x_{shadow}]^T \in \mathbb{R}^d$, where $d = 4$ and compile a tabular dataset. This methodology provides a physics-based dataset where the causal, ground truth causal model is known to show the abilities of CSHTEST and CSVHTEST.

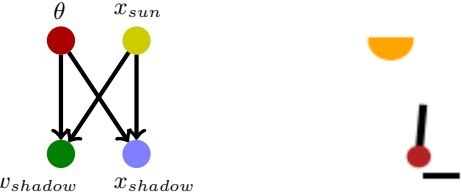

Figure 8: Pendulum toy example. (a) The structural DAG dictating the pendulum tabular dataset. (b) A visual representation of one of the results from the pendulum dataset.

Each pendulum entry is determined by the two exogenous variables, pendulum angle ($\theta$) and sun position ($x_{sun}$). Here, the data samples are generated roughly where the angle of the pendulum and the angle of light from the sun range $\in (-45, 45)$ degrees, generated independently. Then from that, we calculate a physics-based interpretation of the shadow position and width. In the calculation of both of the endogenous variables, we introduce non-linearities in operating on various trigonometric

functions. In the shadow width case, we also deal with an maximum as we don't want the width to go below 0. Afterward, in most of the datasets unless otherwise mentioned, we add Gaussian noise to the endogenous variables in the dataset so that the SNR is 10dB.

### A.2.2   Pendulum Hypotheses

We introduce 6 enumerated hypotheses for the pendulum dataset, enumerated in Figure 9. We have two hypotheses that are 1 Structural Hamming Distance away from ground truth (*leaky* and *lossy*), and 3 that are 2 Structural Hamming Distances away (*2leaky*, *2lossy*, and *leak-loss*). The choice of which edge to add or remove were arbitrary, unless required by design. The individual names of the hypotheses give away their purpose, as they are meant to be leaky or lossy in a specific way to observe the empirical qualities of the different hypothesis tests. Two points of interest. Despite including an additional leakage in *2leaky*, we maintain the exogenousity of $x_{sun}$, meaning that the $\theta$ to $x_{sun}$ is purely a leakage term from the SCM's perspective. Secondly, the *leak-loss* hypothesis has a net Hamming distance of 0 and structurally still has the same connectivity as ground truth. However, due to the choice of paths, it likely has some functional limitations.

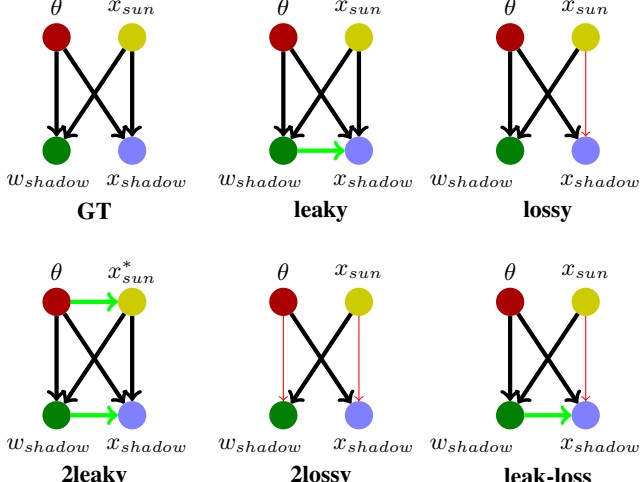

Figure 9: The 6 enumerated pendulum hypotheses that we try out. Red and thin arrows are arrows that we remove from the true DAG and green arrows are arrows that we add to the true DAG. In the case of 2leaky, we maintain $x_{sun}$ as an exogenous variable, but allow $\theta$ information to also influence its value.

We also tried a couple other hypotheses, such as adding the link from $\theta$ to $x_{sun}$ by dropping the exogenousity of $x_{sun}$ and the inverse DAG (which is just the same ground truth DAG except with inverted arrows). Neither of them performed well, so we removed them from further analysis.

### A.2.3   Pendulum Training Architecture and Hyperparameters

Unfortunately, there is a limitation with regards to hyperparameters. Because we are effectively using the average loss over several random initializations onto the generalization set as the primary proxy of SCM "goodness," the hyperparameters that we choose to represent the $\eta$ neural networks do have possible effects in our overall methods.

- 50 Epochs. 40 Iterations.
- Causal $\eta$ networks: $[4, 16, 4]$ nodes per output.
- For CSVHTEST, Encoder and Decoder networks: $[4, 4]$ node MLP per input.
- Normally-distributed initializations.
- Activation: Soft Leaky ReLU.
- Optimizer: PSGD (discussed further in A.4). Initial Learning Rate of 0.01.
- Cosine Annealing Scheduler. Warm Restarts implemented for non-variational models.

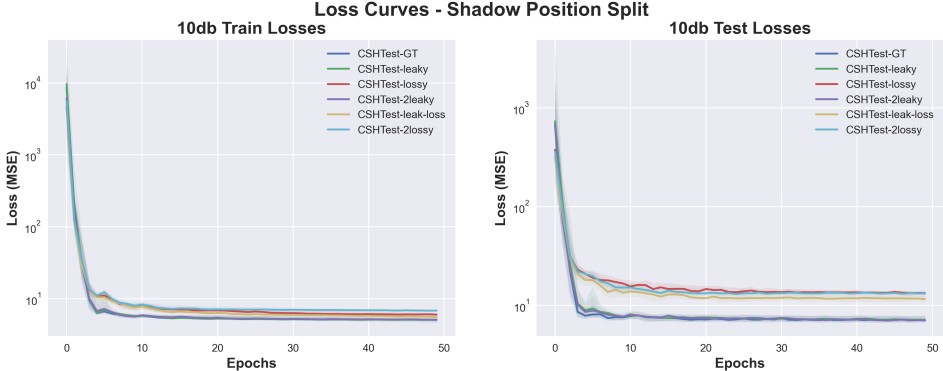

Figure 10: Loss trajectory of the Sun Split OOD Run

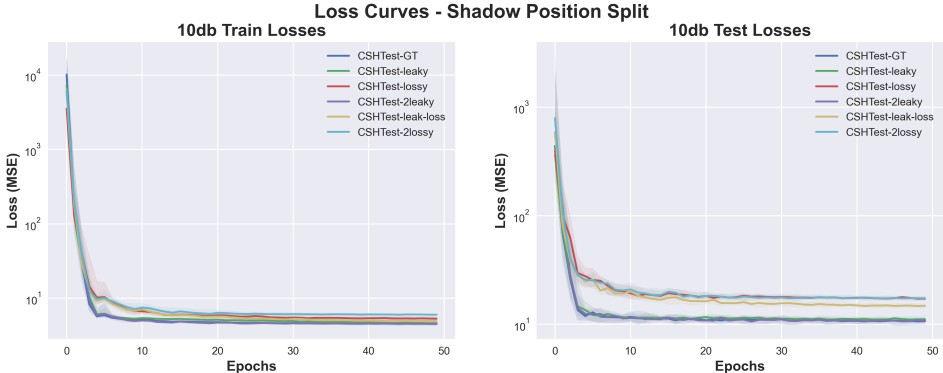

Figure 11: Loss trajectory of the Shadow Position Split OOD Run

### A.2.4 Further Pendulum Results

Full numerical results of Figure 2. Train and Loss trajectory curves are shown in Figures 10 and 11.

| Hypothesis | GT | lossy | leaky | 2lossy | 2leaky | leak-loss |
|---|---|---|---|---|---|---|
| Sun | $7.14 \pm 0.92$ | $13.48 \pm 1.06$ | $7.41 \pm 0.98$ | $13.26 \pm 0.92$ | $7.21 \pm 0.79$ | $11.66 \pm 1.15$ |
| Shadow Position | $10.63 \pm 1.09$ | $17.29 \pm 1.16$ | $11.02 \pm 0.93$ | $17.56 \pm 0.60$ | $10.98 \pm 2.16$ | $14.75 \pm 1.08$ |

Table 2: Mean and Standard Deviation of the Final Test Loss in Pendulum Hypothesis Testing Experiments.

## A.3 Background Theory

### A.3.1 Variational Hypothesis testing and Data Generation with CSVHTEST

We extend CSHTEST to a variational model CSVHTEST, that includes sampling functionality like a VAE [2]. Thus CSVHTEST can generate new data points that are not deterministic on the inputs, allowing for synthetic data generation. CSVHTEST consists of an encoder, a CSHTEST causal layer and a decoder.

These encoder and decoder networks do not compress the data, but enable a transformation of the inputs to a normally distributed space, enabling sampling without preventing the relevance of the causal priors given by $\mathbf{S}_i$. Thus, CSVHTEST is an extension of CSHTEST with

$$\boldsymbol{z}_i = f_{enc}(\boldsymbol{x}_i), \quad \hat{\boldsymbol{z}}_i = \eta_i(\mathbf{S}_i \circ \boldsymbol{z}), \quad \hat{\boldsymbol{x}}_i = f_{dec}(\hat{\boldsymbol{z}}_i) \tag{7}$$

The loss function for CSVHTEST includes a weighted Kullback–Leibler (KL) divergence loss to normalize the latent space on top of the reconstruction loss as in CSHTEST. We also add a weighted latent reconstruction loss for the embedded CSHTEST which enforces separation of the encoder and decoders as transformations, and the $\eta$ networks as the functional approximators on these transformations.

$$\ell_{KL} = KL(\boldsymbol{z}_i || \mathcal{N}(0,1)) \tag{8}$$

$$\ell_{latent} = \ell(\boldsymbol{z}, \hat{\boldsymbol{z}}) \tag{9}$$

$$\ell_{MSE} = ||\boldsymbol{x} - \eta_i(\mathbf{S}_i \circ \boldsymbol{x})||_2 \tag{10}$$

$$\ell_{\text{CSHTEST}} = \ell_{MSE} + \lambda_{KL} * \ell_{KL} + \lambda_{latent} * \ell_{latent} \tag{11}$$

### A.3.2 Constructing a Causal Generative Model

Following the classic VAE model, given inputs $\mathbf{x}$, we encode into a latent space $\mathbf{z}$ with distribution $q_\phi$ where we have priors given by $p(\cdot)$ [2].

$$\text{ELBO} = \mathbb{E}_{q_\mathcal{X}} \left[ \mathbb{E}_{\mathbf{z} \sim q_\phi} \left[ \log p_\theta(\mathbf{x}|\mathbf{z}) \right] - \mathcal{D}(q_\phi(\mathbf{z}|\mathbf{x}) || p_\theta(\mathbf{z})) \right] \tag{12}$$

In [6], the causal layer is described as a noisy linear SCM:

$$\mathbf{z} = \mathbf{S}^T \mathbf{z} + \boldsymbol{\epsilon} \tag{13}$$

which finds some causal structure of the latent space variables $\mathbf{z}$ with respect to a matrix $\mathbf{S}$. By itself, $\mathbf{S}$ functions as the closest linear approximator for the causal relationships in the latent space of $\mathbf{z}$.

A non-linear mask is applied to the causal layer so that it can more accurately estimate non-linear situations as well. Suppose $\mathbf{S}$ is composed of column vectors $\mathbf{S}_i$. For each latent space concept $i$, define a non-linear function $g_i : \mathbb{R}^n \to \mathbb{R}$ and modify equation (13) such that

$$\mathbf{z}_i = g_i(\mathbf{S}_i \circ \mathbf{z}) + \boldsymbol{\epsilon} \tag{14}$$

where $\circ$ is the Hadamard product. In this formulation, the view of $\mathbf{S}$ changes from one of function estimation to one of adjacency. That is, if $\mathbf{S}$ is viewed as a binary adjacency matrix, the $g_i$ functions take the responsibility of reconstructing $\mathbf{z}$ given only the the parents, dictated by $\mathbf{S}_i \circ \mathbf{z}$. In the simplest case, if $g_i(\mathbf{v}) = \sum_j v_j$, the summation of all the values of $\mathbf{v}$, then Equation (14) degenerates back to Equation (13) [15].

Including the causal layer introduces many auxiliary loss functions that we mostly adopt [6]. First is a label loss (15), where the adjacency matrix $\mathbf{S}$ should also apply to the labels $\mathbf{u}$. This loss is used in pre-training in its linear form to learn a form of $\mathbf{S}$ prior to learning the encoder and decoders. After pre-training, we apply a nonlinear mask $f_i$ that functions similarly to $g_i$, but operates on the label space directly, but with the same $\mathbf{S}$.

$$\ell_u = \mathbb{E}_{q_\mathcal{X}} \left[ \sum_{i=1}^n ||u_i - f_i(\mathbf{S}_i \circ \mathbf{u})||^2 \right] \tag{15}$$

The latent loss tries to enforce the SCM, described by Equation (14).

$$\ell_z = \mathbb{E}_{\mathbf{z} \sim q_\phi} \left[ \sum_{i=1}^n \left\| z_i - g_i(\mathbf{S_i} \circ \mathbf{z})^2 \right\| \right] \tag{16}$$

Further enforcing the label spaces, we can define a prior $p(\mathbf{z}|\mathbf{u})$. We use the same conventions as in [6] and say that

$$p(\mathbf{z}|\mathbf{u}) \sim \mathcal{N}(\mathbf{u}_n, \mathbf{I})$$

where $\mathbf{u}_n \in [-1, 1]$ are normalized label values. This translates to an additional KL-loss.

We notice that the Variational version CSVHTEST often works better than CSHTEST in the presence of noise. Results are shown in Figure 12.

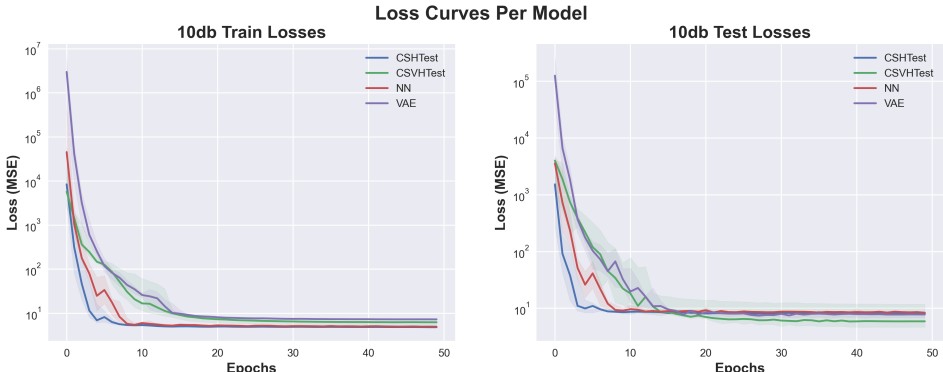

Figure 12: Comparison of CSHTEST, CSVHTEST, NN, and VAE in the presence of noise

## A.4 Causal Problems and the Investigation of Optimizers

While working on the causal problems, because of the input of so many zeros in the structural Hadamard product, the loss space is not very well-behaved. As a result, we notice some inconsistency in loss trajectory with different optimizers. We do an initial investigation on the possibilities of different optimizers in our problem space.

Initially optimizers Adam, SGD, Adabelief, and PSGD [16–19] are compared primarily for mean OOD MSE test loss across iterations in a limited number of test cases [20, 21]. Due to better performance, Adabelief and PSGD are compared in a more robust set of cases. Figure 13 shows a scatter plot of the final losses across all test cases, and a subset of the results are listed in Table A.4. Although PSGD routinely outperform AdaBelief both in mean final loss and variance (across 3 iterations per test), there were select conditions and DAGs in which any single optimizer would underperform or not converge. We leave it to further research to investigate optimizer performance and considerations for causally informed deep learning architecture that are constrained in unique ways than traditional deep learning models. PSGD had better loss in 171 cases, and lower variance in 148 of the 176 test cases.

Optimizer Test Cases (176 total, every combination of below):

- DAG Size (number of nodes, number of edges): (4,4), (5,5)

- SEM: linear, nonlinear generative functions

- Model: CSHTEST, CSVHTEST

- SNR: 0 noise (inf SNR), 7 dB

- Hamm (Sturctural Hamming Distance): 0 (ground truth), 1

- Split: ID (in-dsitribution or random), OOD (out-of-dist. or 75% quantile split)

- Split Number: Which node/variable is data being split on (not relevant to ID)

Table A.4 has results for DAG Size (4,4) nonlinead

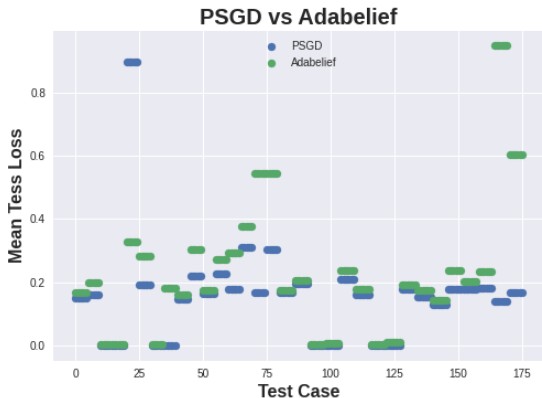

Figure 13: Comparison of final test loss of optimizers PSGD and AdaBelief across 176 unique tests

| Model | SNR | Hamm | Split | Split_Num | AdaBelief | PSGD |
|---|---|---|---|---|---|---|
| CHGen | 7.0 | 0 | ID | 1 | $0.17 \pm 2.26\text{e-}04$ | $0.15 \pm 6.74\text{e-}06$ |
| | | | OOD | 0 | $0.17 \pm 2.26\text{e-}04$ | $0.15 \pm 6.74\text{e-}06$ |
| | | | | 1 | $0.17 \pm 2.26\text{e-}04$ | $0.15 \pm 6.74\text{e-}06$ |
| | | | | 2 | $0.17 \pm 2.26\text{e-}04$ | $0.15 \pm 6.74\text{e-}06$ |
| | | | | 3 | $0.17 \pm 2.26\text{e-}04$ | $0.15 \pm 6.74\text{e-}06$ |
| | | 1 | ID | 1 | $0.2 \pm 6.69\text{e-}03$ | $0.16 \pm 1.18\text{e-}03$ |
| | | | OOD | 0 | $0.2 \pm 6.69\text{e-}03$ | $0.16 \pm 1.18\text{e-}03$ |
| | | | | 1 | $0.2 \pm 6.69\text{e-}03$ | $0.16 \pm 1.18\text{e-}03$ |
| | | | | 2 | $0.2 \pm 6.69\text{e-}03$ | $0.16 \pm 1.18\text{e-}03$ |
| | | | | 3 | $0.2 \pm 6.69\text{e-}03$ | $0.16 \pm 1.18\text{e-}03$ |
| | inf | 0 | ID | 1 | $0.0 \pm 4.07\text{e-}05$ | $0.0 \pm 9.75\text{e-}10$ |
| | | | OOD | 0 | $0.0 \pm 4.07\text{e-}05$ | $0.0 \pm 9.75\text{e-}10$ |
| | | | | 1 | $0.0 \pm 4.07\text{e-}05$ | $0.0 \pm 9.75\text{e-}10$ |
| | | | | 2 | $0.0 \pm 4.07\text{e-}05$ | $0.0 \pm 9.75\text{e-}10$ |
| | | | | 3 | $0.0 \pm 4.07\text{e-}05$ | $0.0 \pm 9.75\text{e-}10$ |
| | | 1 | ID | 1 | $0.0 \pm 3.92\text{e-}05$ | $0.0 \pm 2.54\text{e-}10$ |
| | | | OOD | 0 | $0.0 \pm 3.92\text{e-}05$ | $0.0 \pm 2.54\text{e-}10$ |
| | | | | 1 | $0.0 \pm 3.92\text{e-}05$ | $0.0 \pm 2.54\text{e-}10$ |
| | | | | 2 | $0.0 \pm 3.92\text{e-}05$ | $0.0 \pm 2.54\text{e-}10$ |
| | | | | 3 | $0.0 \pm 3.92\text{e-}05$ | $0.0 \pm 2.54\text{e-}10$ |
| CVHGen | 7.0 | 0 | ID | 1 | $0.33 \pm 4.85\text{e-}02$ | $0.9 \pm 1.54\text{e+}00$ |
| | | | OOD | 0 | $0.33 \pm 4.85\text{e-}02$ | $0.9 \pm 1.54\text{e+}00$ |
| | | | | 1 | $0.33 \pm 4.85\text{e-}02$ | $0.9 \pm 1.54\text{e+}00$ |
| | | | | 2 | $0.33 \pm 4.85\text{e-}02$ | $0.9 \pm 1.54\text{e+}00$ |
| | | | | 3 | $0.33 \pm 4.85\text{e-}02$ | $0.9 \pm 1.54\text{e+}00$ |
| | | 1 | ID | 1 | $0.28 \pm 2.72\text{e-}02$ | $0.19 \pm 9.08\text{e-}04$ |
| | | | OOD | 0 | $0.28 \pm 2.72\text{e-}02$ | $0.19 \pm 9.08\text{e-}04$ |
| | | | | 1 | $0.28 \pm 2.72\text{e-}02$ | $0.19 \pm 9.08\text{e-}04$ |
| | | | | 2 | $0.28 \pm 2.72\text{e-}02$ | $0.19 \pm 9.08\text{e-}04$ |
| | | | | 3 | $0.28 \pm 2.72\text{e-}02$ | $0.19 \pm 9.08\text{e-}04$ |
| | inf | 0 | ID | 1 | $0.0 \pm 3.55\text{e-}06$ | $0.0 \pm 1.08\text{e-}09$ |
| | | | OOD | 0 | $0.0 \pm 3.55\text{e-}06$ | $0.0 \pm 1.08\text{e-}09$ |
| | | | | 1 | $0.0 \pm 3.55\text{e-}06$ | $0.0 \pm 1.08\text{e-}09$ |
| | | | | 2 | $0.0 \pm 3.55\text{e-}06$ | $0.0 \pm 1.08\text{e-}09$ |
| | | | | 3 | $0.0 \pm 3.55\text{e-}06$ | $0.0 \pm 1.08\text{e-}09$ |
| | | 1 | ID | 1 | $0.18 \pm 8.86\text{e-}02$ | $0.0 \pm 1.62\text{e-}08$ |
| | | | OOD | 0 | $0.18 \pm 8.86\text{e-}02$ | $0.0 \pm 1.62\text{e-}08$ |
| | | | | 1 | $0.18 \pm 8.86\text{e-}02$ | $0.0 \pm 1.62\text{e-}08$ |
| | | | | 2 | $0.18 \pm 8.86\text{e-}02$ | $0.0 \pm 1.62\text{e-}08$ |
| | | | | 3 | $0.18 \pm 8.86\text{e-}02$ | $0.0 \pm 1.62\text{e-}08$ |

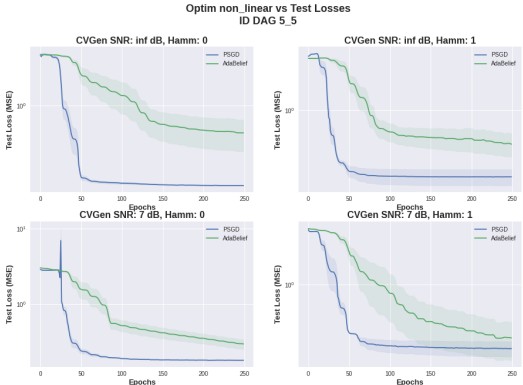

Figure 14: Example of loss curves for optimizers PSGD and AdaBelief inf,1 dB SNR and 0,1 Hamming Distance.

## A.5  DAG Simulation Settings

### A.5.1  Training Hyperparameters

The following fixed setting where used when training models for the simulations. A random seed (1) was used in all experiments.

- 100 Epochs

- Random Weight Matrix $\mathbf{N}(0, 1)$ for linear model weights

- Linear model $\eta$ nets size: [4,4] MLP

- Non-linear model $\eta$ nets size: [4,16,8,2] MLP

- activation function: Soft Leaky ReLU

- 10 Iterations of a Ground Truth DAG per DAG size

- 5 Iterations of modified DAGs per Hamming Distance

- 100 data points (N) per DAG of training data

- Optimzer: PSGD

- Noise Gaussian, 0-mean, variance calculated per SNR

- Split Number: Which node/variable is data being split on (not relevant to ID)

### A.5.2  Test Cases

- DAG Size (number of nodes, number of edges): (4,4), (5,5)

- SEM: linear generative functions

- Model: CSHTEST

- SNR: 0 noise (inf SNR), 5 dB

- Hamm (Structural Hamming Distance): 0 (ground truth), 1, 2, 3, 4

- OOD (out-of-dist. or 75% quantile split)

- Split Number: Which node/variable is data being split, one per each variable (4 for a 4 node graph)

## A.6 Final Losses with Sample Variances

| Hypothesis | Split | Model | Train | Test |
|---|---|---|---|---|
| H1 | Random | CGen | $0.02 \pm 2.34e - 07$ | $0.02 \pm 9.42e - 08$ |
| | | CVGen | $7.09 \pm 1.71e + 00$ | $8.21 \pm 2.44e + 00$ |
| | DBP 75% | CGen | $0.02 \pm 2.00e - 06$ | $0.04 \pm 3.82e - 06$ |
| | | CVGen | $0.74 \pm 1.81e - 02$ | $0.23 \pm 2.32e - 02$ |
| | GCS 25% | CGen | $0.02 \pm 4.67e - 07$ | $0.03 \pm 2.67e - 06$ |
| | | CVGen | $0.39 \pm 4.62e + 00$ | $1.08 \pm 4.28e + 00$ |
| H2 | Random | CGen | $0.02 \pm 6.01e - 08$ | $0.02 \pm 3.88e - 07$ |
| | | CVGen | $7.1 \pm 3.82e + 00$ | $8.17 \pm 5.36e + 00$ |
| | DBP 75% | CGen | $0.02 \pm 5.07e - 07$ | $0.03 \pm 6.59e - 08$ |
| | | CVGen | $0.74 \pm 3.66e - 03$ | $0.17 \pm 3.16e + 01$ |
| | GCS 25% | CGen | $0.02 \pm 1.30e - 07$ | $0.06 \pm 5.45e - 05$ |
| | | CVGen | $0.36 \pm 2.44e + 00$ | $3.83 \pm 3.94e + 00$ |

