# OpenReview forum: "Causal Structural Hypothesis Testing and Data Generation Models"
_NeurIPS.cc/2022/Workshop/SyntheticData4ML — Neurips 2022 SyntheticData4ML_

### Official Review · Reviewer_Pdqi · 2022-10-16
**Interesting idea but connections to related work should be clarified more**

**Rating:** 6
**Confidence:** 3

**Review:**

# Summary
The authors propose two generative models that assume access to a causal DAG to learn the structural assignment functions. They then apply this for hypothesis testing and out-of-distribution data generation.

# Strengths
* the causal hypothesis testing problem is well-motivated as causal DAG assumptions are commonly made/available in many real-world sciences
* the experimental settings are very interesting and meaningful

# Suggestions
* there already exist various methods that assume access to a causal DAG and learn the structural equations only. See Section 4.1 in [1] for a detailed comparison. It would be good if the authors could clarify how their generative models stand out compared to these methods and empirically verify their superiority
* it is unclear when to use CGen over CVGen or vice versa


[1] Causal Machine Learning: A Survey and Open Problems, Jean Kaddour, Aengus Lynch, Qi Liu, Matt J. Kusner, Ricardo Silva, https://arxiv.org/abs/2206.15475

---

### Official Review · Reviewer_6WYM · 2022-10-18
**Interesting new approach---however lacks comparable SCM validation baselines and error bars to be convincing**

**Rating:** 4
**Confidence:** 3

**Review:**

Summary: This paper provides a new approach to perform structural causal model hypothesis testing by training a generative model on an informed train-test split of the collected dataset for a causal task.


Strengths and weaknesses:

Originality: This paper appears to propose two novel generative model approaches, however causal generative models are outside my domain of expertise to comment on whether their approach is novel, and as such I will leave that for other more informed reviewers. One weakness is that the field of hypothesis testing to validate a causal model seems to be an established field [1,2], and none of this literature was discussed in the brief related work, and I would expect to see comparisons against the best methods to validate from a set of multiple causal models, instead of meaningless baseline methods of training a generative model of a neural network or a VAE.

Quality: The method in section 3 is clear mathematically, however the claims are not well supported by the experimental results, as there is no comparison to baseline SCM hypothesis testing methods, no guidelines when we should reject the other SCM models, e.g. what happens if two different SCM models have the same lowest train and test MSE error? Or overlapping standard deviation errors? Furthermore, the experimental results are limiting, as they seem to not have been repeated for any random seeds, therefore it is not possible to judge whether the method does significantly identify the true SCM or do the SCM have overlapping standard deviation MSE errors?

Clarity: The paper is poorly written, poorly organised when referencing tables or experimental results. This can be improved by improving the flow of the paper, correctly situating the problem and the related work, when referring to a Figure (Table), prepend the word Figure (Table) if referring to a figure, e.g. we observe this in Figure 5. It would also help the reader to have a short sentence or paragraph at the beginning of each section outlining what is to be expected, e.g. In this section we outline XX and observe YY, then show the property of Z in X. The paper would benefit from a proof-read, and or even a word processor to check grammar---there are many sentences in the paper that are wrong or incomplete, e.g. Section titles 3.1 and 3.2 "Causal Hypothesis testing with", ending a sentence using the word "with" is an incomplete sentence and should be completed by appending another word. Furthermore a proof read would pick up miss-labelled words, such as line 68 "network-network" should be "neural-network" etc. Also sub-figures should be correctly labelled, i.e. have (a) in the sub-figure when referring to (a) in the sub-figure in the text.

Significance: The paper proposes an interesting approach, however it is difficult to assess whether they are meaningful due to the lack of error bars in their results. The approach does seem to be useful for the problem of SCM validation, which in itself has useful applications.

It would instructive for the reader to include known limitations with the proposed method as well.

Questions: How does this paper fit into the related work of SCM validation [1,2]?
Is it possible to re-do the experiments over 10-20 random seeds to produce error bars for the results?
Is it possible to re-write the paper such that it is clearer and flows better ? Some helpful resources of a good sample paper [3] and ICML recommendations for writing [4]

A paper which clears these concerns would clear the bar, however I feel that it does not in its current state.

References:

[1] http://www.its.caltech.edu/~fehardt/UAI2016WS/papers/Tran.pdf
[2] https://arxiv.org/pdf/1711.08936v2.pdf
[3] https://arxiv.org/pdf/1703.10593.pdf
[4] https://icml.cc/Conferences/2002/craft.html

---

### Official Review · Reviewer_KL4o · 2022-10-20
**Interesting paper; a few questions**

**Rating:** 6
**Confidence:** 3

**Review:**

I think this paper introduces some interesting concepts. The idea of hypothesis testing with variational models seems interesting and a valuable approach to the problem. Furthermore, the presented ideas seem to be confirmed in experiments (though I believe there should be a few more benchmarks beyond a VAE and NN).

There are however a few questions I think a full version of this paper should answer:

* It is not clear to me _which_ hypotheses are tested. Are we confirming/rejecting the identifiability of the causal edge? If so, there are many proofs that show this is impossible without additional assumptions. This is fine of course, but with highly flexible variational models, I don't see how these assumptions are encoded here (note that It is possible I am missing some crucial points of the paper here).

* How is the hypothesis tested? The methods section seems to focus mainly on data generation; are we sampling synthetic data and comparing it with the real data? If the latter is the case, we can not really test causal hypotheses as even from a non-causal model we can generate highly believable data.


Overall I think the paper is interesting, mainly due to the diversity of experiments, but it needs some serious work in explaining the method/hypotheses in more detail.

---

### Meta-Review · Area_Chair_sd3H · 2022-10-19

**Recommendation:** Accept